# GRAND: Graph Neural Diffusion

**Benjamin P. Chamberlain**[*]  **James Rowbottom**[*]  **Maria Gorinova**  **Stefan Webb**
Twitter Inc.                    Twitter Inc.                Twitter Inc.            Facebook Inc.
bchamberlain@twitter.com

**Emanuele Rossi**                              **Michael M. Bronstein**
Twitter Inc. and Imperial College London        Twitter Inc. and Imperial College London

## Abstract

We present Graph Neural Diffusion (GRAND), a model that approaches deep learning on graphs as a continuous diffusion process and treats Graph Neural Networks (GNNs) as discretisations of an underlying PDE. In our model, the layer structure and topology correspond to the discretisation choices of temporal and spatial operators. Our approach allows a principled development of a broad new class of GNNs that are able to address the common plights of graph learning models such as depth, oversmoothing, and bottlenecks. Key to the success of our models are stability with respect to perturbations in the data and this is addressed for both implicit and explicit discretisation schemes. We develop linear and nonlinear versions of GRAND, achieving competitive results on many standard graph benchmarks.

## 1 Introduction

Graph Neural Networks (GNNs) are intimately connected to differential equations. The seminal work of [12] was concerned with finding the fixed points of differential equations using the Almeida-Pineda algorithm [1, 11]. The currently predominant message passing paradigm [6] can be modelled as a differential equation. More recently, diffusion processes have been shown to be an effective preprocessing step for graph learning [7]. PDEs are among the most studied mathematical constructions, and historically, PDE-based methods have been used extensively in signal and image processing [10], computer graphics [13], and machine learning [5, 4].

Our goal is to show that the tools of PDEs can be used to understand existing GNN architectures and as a principled way to develop a broad class of new methods. We focus on architectures that can be interpreted as information diffusion on graphs, modelled by the diffusion equation. In doing so, we show that many popular GNNs can be derived from a single mathematical framework by different choices of the form of diffusion equation and discretisation schemes. Standard GNNs are equivalent to the explicit single-step Euler scheme that is inefficient and requires small step sizes. We show that more advanced, adaptive multi-step schemes such as Runge-Kutta perform significantly better and using implicit schemes, which are unconditionally stable, amounts to larger multi-hop diffusion operators. Choosing different spatial discretisation amounts to *graph rewiring*, a technique recently used to improve the performance of GNNs [7, 2]. We show that appropriate choices within our framework allow the design of deep GNN architectures with tens of layers. This is a feat hard to achieve otherwise due to feature oversmoothing [8, 9] and bottlenecks [2] – phenomena that are recognised as a common plight of most graph learning architectures.

## 2 Background

**Diffusion equation**  Let $x(t)$ denote a family of scalar-valued functions on $\Omega \times [0, \infty)$ representing the distribution of some property (which we will assume to be temperature for simplicity) on a domain

35th Conference on Neural Information Processing Systems (NeurIPS 2021), Sydney, Australia.

$\Omega$ at some time, and let $x(u, t)$ be its value at point $u \in \Omega$ at time $t$. According to Fourier's law of heat conduction, the heat flux $h = -g\nabla x$, is proportional to the temperature *gradient* $\nabla x$, where $g$ is the *diffusivity* describing the thermal conductance properties of $\Omega$. An idealized *homogeneous* setting assumes that $g$ is a constant scalar throughout $\Omega$. More generally, the diffusivity is a *inhomogeneous* (position-dependent) function that can be scalar-valued (in which case it simply scales the temperature gradient and is *isotropic*) or matrix-valued (in which case the diffusion is said to be *anisotropic*, or direction-dependent). The continuity condition $x_t = -\text{div}(h)$ (roughly meaning that the only change in the temperature is due to the heat flux, as measured by the *divergence* operator, i.e., heat is not created or destroyed), leads to a PDE referred to as the *(heat) diffusion equation*,

$$\frac{\partial x(u,t)}{\partial t} = \text{div}[g(u, x(u,t), t)\nabla x(u,t)],$$

with the initial condition $x(u, 0) = x_0(t)$; for simplicity, we assume no boundary conditions. The choice of the diffusivity function determines if the diffusion is homogeneous ($g = c$), inhomogeneous ($g(u,t)$), or anisotropic ($A(u,t)$). In the isotropic case, the diffusion equation can be expressed as

$$\frac{\partial x(u,t)}{\partial t} = \text{div}(c\nabla x) = c\Delta x, \tag{1}$$

where $\Delta x = \text{div}(\nabla x)$ is the *Laplacian* operator.

**Diffusion on manifolds**   In our discussion so far we assumed some abstract domain $\Omega$. The structure of the domain is manifested in the definition of the spatial differential operators in the diffusion PDE. In a general setting, we model $\Omega$ as a Riemannian manifold, and let $\mathcal{X}(\Omega)$ and $\mathcal{X}(T\Omega)$ denote the spaces of *scalar* and *(tangent) vector fields* on it, respectively. We denote by $\langle x, y \rangle$ and $\langle\!\langle \mathscr{X}, \mathscr{Y} \rangle\!\rangle$ the respective inner products on $\mathcal{X}(\Omega)$ and $\mathcal{X}(T\Omega)$. Furthermore, we denote by $\nabla : \mathcal{X}(\Omega) \to \mathcal{X}(T\Omega)$ and $\text{div} = \nabla^* : \mathcal{X}(T\Omega) \to \mathcal{X}(\Omega)$ the *gradient* and *divergence* operators, which are adjoint w.r.t. the above inner products: $\langle\!\langle \nabla x, \mathscr{X} \rangle\!\rangle = \langle x, \text{div}(\mathscr{X}) \rangle$. Informally, the gradient $\nabla x$ of a scalar field $x$ is a vector field providing at each point $u \in \Omega$ the direction $\nabla x(u)$ of the steepest change of $x$. The divergence $\text{div}(\mathscr{X})$ of a vector field $\mathscr{X}$ is a scalar field providing, at each point, the flow of $\mathscr{X}$ through an infinitesimal volume. The Laplacian $\Delta x$ can be interpreted as the local difference between the value of a scalar field $x$ at a point and its infinitesimal neighbourhood.

## 3   Diffusion equations on graphs

We now define *diffusion equations on graphs*, analogous to Section 2 and argue that formalizing GNNs under the diffusion equation framework provides a principled and rigorous way to develop new architectures for graph learning.

### 3.1   Graph diffusion equation

Let $\mathcal{G} = (\mathcal{V}, \mathcal{E})$ be an undirected graph with $|\mathcal{V}| = n$ nodes and $|\mathcal{E}| = e$ edges, and let $\mathbf{x}$ and $\mathscr{X}$ denote features defined on nodes and edges respectively.[1] The node and edge fields can be represented as $n$- and $e$-dimensional vectors assuming some arbitrary ordering of nodes. We adopt the same notation for the respective inner products:

$$\langle \mathbf{x}, \mathbf{y} \rangle = \sum_{i \in \mathcal{V}} x_i y_i \qquad \langle\!\langle \mathscr{X}, \mathscr{Y} \rangle\!\rangle = \sum_{i > j} w_{ij} \mathscr{X}_{ij} \mathscr{Y}_{ij}$$

Here, $w_{ij}$ denotes the *adjacency* of $\mathcal{G}$: $w_{ij} = w_{ji} = 1$ iff $(i, j) \in \mathcal{E}$. We tacitly assume edge fields to be *alternating*, so $\mathscr{X}_{ji} = -\mathscr{X}_{ij}$, and no self-edges, so $(i, i) \notin \mathcal{E}$. The gradient $(\nabla\mathbf{x})_{ij} = x_j - x_i$ assigns the edge $(i, j) \in \mathcal{E}$ the difference of its endpoint features and is alternating by definition. Similarly, the divergence $(\text{div}(\mathscr{X}))_i$ assigns the node $i$ the sum of the features of all edges it shares:

$$(\text{div}(\mathscr{X}))_i = \sum_{j:(i,j)\in\mathcal{E}} \mathscr{X}_{ij} = \sum_{j=1}^n w_{ij} \mathscr{X}_{ij}$$

---

[1]For simplicity, we assume these features to be scalar-valued and refer to them as node and edge fields, by analogy to scalar and vector fields on manifolds. In the rest of the paper, we assume vector-valued node features, a straightforward extension.

The two operators are adjoint, $\langle\!\langle \nabla\mathbf{x}, \mathcal{X} \rangle\!\rangle = \langle \mathbf{x}, \text{div}(\mathcal{X}) \rangle$. We consider the following diffusion equation on the graph

$$\frac{\partial\mathbf{x}(t)}{\partial t} = \text{div}[\mathbf{G}(\mathbf{x}(t),t)\nabla\mathbf{x}(t)] \tag{2}$$

with an initial condition $\mathbf{x}(0)$. Here we denote by $\mathbf{G} = \text{diag}(a(x_i(t), x_j(t), t))$ an $e \times e$ diagonal matrix and $a$ is some function determining the similarity between nodes $i$ and $j$. While in general $a(x_i, x_j, t)$ can be time-dependent, we will assume $a = a(x_i, x_j)$ for the sake of simplicity. Plugging in the expressions of $\nabla$ and $\text{div}$, we get

$$\frac{\partial}{\partial t}\mathbf{x}(t) = (\mathbf{A}(\mathbf{x}(t)) - \mathbf{I})\mathbf{x}(t) = \bar{\mathbf{A}}(\mathbf{x}(t))\mathbf{x}(t) \tag{3}$$

where $\mathbf{A}(\mathbf{x}) = (a(x_i, x_j))$ is the $n \times n$ *attention matrix* with the same structure as the adjacency of the graph (we assume $a_{ij} = 0$ if $(i,j) \notin \mathcal{E}$). Note that in the setting when $\mathbf{A}(\mathbf{x}(t)) = \mathbf{A}$ we get a linear diffusion equation that can be solved analytically as $\mathbf{x}(t) = e^{\bar{\mathbf{A}}t}\mathbf{x}(0)$.

## 3.2 Properties of the graph diffusion equation

Differential equation stability is closely related to the concept of robustness in machine learning; changes in model outputs should be small under small changes in inputs. Formally, a solution $\mathbf{x}(t)$ of the PDE is said to be stable, if given any $\epsilon > 0$ there exists $\delta > 0$ such that for any solution $\hat{\mathbf{x}}(t)$, such that $|\mathbf{x}(0) - \hat{\mathbf{x}}(0)| \leq \delta$, it is also the case that $|\mathbf{x}(t) - \hat{\mathbf{x}}(t)| \leq \epsilon$ for all $t \geq 0$.

In the linear case, it is sufficient to show that the eigenvalues of $\bar{\mathbf{A}}$ are non-positive (see the Supplementary Materials for proof). For the general nonlinear case, we show

$$\max_i x_i(0) \geq x_i(t) \geq \min_i x_i(0) \quad \forall t \geq 0, \tag{4}$$

which follows from (i) the function $\bar{A}(\mathbf{x})\mathbf{x}$ being continuous in $\mathbf{x}$, (ii) the largest component of $\mathbf{x}(t)$ not increasing in time, and (iii) the smallest component is not decreasing in time.

Condition (i) holds as $\bar{\mathbf{A}}$ is a composition of Lipschitz-continuous functions (cf. equation (6)). Defining indices $k = \arg\max_i x_i$ and $l = \arg\min_i x_i$ we have

$$\frac{\partial x_k}{\partial t} = \sum_j \bar{a}_{kj}(x)x_j \leq x_k \sum_j \bar{a}_{kj} = 0, \quad \frac{\partial x_l}{\partial t} = \sum_j \bar{a}_{lj}(x)x_j \geq x_l \sum_j \bar{a}_{lj} = 0 \tag{5}$$

since $\mathbf{A}$ is right stochastic, which proves (ii) and (iii). Furthermore, the derivative $\frac{\partial}{\partial x}\mathbf{A}(\mathbf{x})$ is Lipschitz-continuous (from the definition of the attention function we use). Taken together with continuity in time, the requirements of Picard-Lindelöf are satisfied and the PDE is also well posed.

## 4 Graph Neural Diffusion

We now describe Graph Neural Diffusion (GRAND), a new class of GNN architectures derived from the graph diffusion formalism. We assume a given graph $\mathcal{G} = (\mathcal{V}, \mathcal{E})$ with $n$ nodes and $d$-dimensional node-wise features represented as a matrix $\mathbf{X}_{\text{in}}$. GRAND architectures implement the learnable encoder/decoder functions $\phi, \psi$ and a learnable graph diffusion process, to produce node embeddings $\mathbf{Y} = \psi(\mathbf{X}(T))$,

$$\mathbf{X}(T) = \mathbf{X}(0) + \int_0^T \frac{\partial\mathbf{X}(t)}{\partial t}dt, \qquad \mathbf{X}(0) = \phi(\mathbf{X}_{\text{in}})$$

$\frac{\partial\mathbf{X}(t)}{\partial t}$ is given by the graph diffusion equation (2). Different GRAND architectures amount to the choice of the learnable diffusivity function $\mathbf{G}$ and spatial/temporal discretisations of equation (2).

The diffusivity is modelled with an attention function $a(.,.)$. Empirically, scaled dot product attention [14] outperforms the Bahdanau [3] attention used in GAT [15]. The scaled dot product attention is given by

$$a(\mathbf{X}_i, \mathbf{X}_j) = \text{softmax}\left(\frac{(\mathbf{W}_K\mathbf{X}_i)^\top\mathbf{W}_Q\mathbf{X}_j}{d_k}\right), \tag{6}$$

where $\mathbf{W}_K$ and $\mathbf{W}_Q$ are learned matrices, and $d_k$ is a hyperparameter determining the dimension of $W_k$. We use multi-head attention which is useful to stabilise the learning [15, 14] by taking the expectation, $\mathbf{A}(\mathbf{X}) = \frac{1}{h}\sum_h \mathbf{A}^h(\mathbf{X})$. The attention weight matrix $\mathbf{A} = \big(a(\mathbf{X}_i, \mathbf{X}_j)\big)$ is right-stochastic, allowing equation (8) to be written as

$$\frac{\partial}{\partial t}\mathbf{X} = (\mathbf{A}(\mathbf{X}) - \mathbf{I})\mathbf{X} = \bar{\mathbf{A}}(\mathbf{X})\mathbf{X} \tag{7}$$

A broad range of discretisations are possible. Temporal discretisations amount to the choice of numerical scheme, which can use either fixed or adaptive step sizes and be either explicit or implicit. Time forms a continuous analogy to the layer index, where each *layer* corresponds to an iteration of the solver. When using adaptive time step solvers, the number of layers is not specified a-priori. Explicit schemes invoke residual structures that are usually more complex than those employed in standard resnets and which follow directly from rigorous numerical stability results. Implicit numerical schemes offer a natural way of trading off *depth* and *width* (spatial support of the diffusion kernel).

Spatial discretisation amounts to modifying the given graph, or building one in settings where no graph is given and the data can be assumed to lie in some feature space or on a continuous manifold. When the input graph is given, we can rewire the given graph and use a different edge set in the diffusion equation.

While in general equation (7) is nonlinear due to the dependence of $\mathbf{A}$ on $\mathbf{X}$, it becomes linear if the attention weights are fixed inside the integral, $\bar{\mathbf{A}}(\mathbf{X}(t)) = \bar{\mathbf{A}}$ (note that $\mathbf{A}$ is still parametric and learnable, but does not change throughout the diffusion process). In this case, equation (7) can be solved analytically as $\mathbf{X}(t) = e^{\bar{\mathbf{A}}t}\mathbf{X}(0)$. As $\bar{\mathbf{A}}$ is a form of normalised Laplacian, all eigenvalues are non-positive and the steady state solution is given by the dominating eigenvector, which is the degree vector. However, as $\bar{\mathbf{A}}$ is learned, this limitation is not severe as the system can be (and in practice is) degenerate; the graph becomes (approximately) disconnected, with connected components permitted to have unique steady state solutions. We call this model **GRAND-l** for linear to distinguish it from the more general **GRAND-nl** for non-linear. The final variant is **GRAND-nl-rw** (non-linear with rewiring), where rewiring is performed via a two step process: as a preprocessing step, the graph is densified using diffusion weights as in [7], and then at runtime the subset of edges to use is learned based on attention weights. With explicit Euler, Equation (2) becomes:

$$\frac{\mathbf{X}_i^{(k+1)} - \mathbf{X}_i^{(k)}}{\tau} = \sum_{j:(i,j)\in\mathcal{E}'} a\left(\mathbf{X}_i(t), \mathbf{X}_j(t)\right)\left(\mathbf{X}_j(t) - \mathbf{X}_i(t)\right) \tag{8}$$

where $\mathcal{E}' = \{(i,j) : (i,j) \in \mathcal{E} \text{ and } a_{ij} > \rho\}$ with some threshold value $\rho$, is the 'rewired' edge set, which may now contain self-loops. While $a$ changes throughout the diffusion process, rewiring is only performed at the start of the epoch based on features at $t = 0$.

GRAND shares parameters across layer/iteration and is thus more data-efficient than conventional GNNs.

| Random splits | CORA | CiteSeer | PubMed | Coathor CS | Computer | Photo | ogb-arxiv[*] |
|---|---|---|---|---|---|---|---|
| GCN | $81.5 \pm 1.3$ | $\mathbf{71.9 \pm 1.9}$ | $77.8 \pm 2.9$ | $91.1 \pm 0.5$ | $82.6 \pm 2.4$ | $91.2 \pm 1.2$ | $72.17 \pm 0.33$ |
| GAT | $81.8 \pm 1.3$ | $71.4 \pm 1.9$ | $\mathbf{78.7 \pm 2.3}$ | $90.5 \pm 0.6$ | $78.0 \pm 19.0$ | $85.7 \pm 20.3$ | $73.65 \pm 0.11^{\dagger}$ |
| GAT-ppr | $81.6 \pm 0.3$ | $68.5 \pm 0.2$ | $76.7 \pm 0.3$ | $91.3 \pm 0.1$ | $\mathbf{85.4 \pm 0.3}$ | $90.9 \pm 0.3$ | N/A |
| MoNet | $81.3 \pm 1.3$ | $71.2 \pm 2.0$ | $\mathbf{78.6 \pm 2.3}$ | $90.8 \pm 0.6$ | $83.5 \pm 2.2$ | $91.2 \pm 2.3$ | N/A |
| GS-mean | $79.2 \pm 7.7$ | $71.6 \pm 1.9$ | $77.4 \pm 2.2$ | $91.3 \pm 2.4$ | $82.4 \pm 1.8$ | $91.4 \pm 1.3$ | $71.39 \pm 0.16$ |
| GS-maxpool | $76.6 \pm 1.9$ | $67.5 \pm 2.3$ | $76.1 \pm 2.3$ | $85.0 \pm 1.1$ | N/A | $90.4 \pm 1.3$ | N/A |
| CGNN | $81.4 \pm 1.6$ | $66.9 \pm 1.8$ | $66.6 \pm 4.4$ | $\mathbf{92.3 \pm 0.2}$ | $80.29 \pm 2.0$ | $91.39 \pm 1.5$ | $58.70 \pm 2.5$ |
| GDE | $78.7 \pm 2.2$ | $71.8 \pm 1.1$ | $73.9 \pm 3.7$ | $91.6 \pm 0.1$ | $82.9 \pm 0.6$ | $\mathbf{92.4 \pm 2.0}$ | $56.66 \pm 10.9$ |
| **GRAND-l** (ours) | $\mathbf{83.6 \pm 1.0}$ | $\mathbf{73.4 \pm 0.5}$ | $\mathbf{78.8 \pm 1.7}$ | $\mathbf{92.9 \pm 0.4}$ | $83.7 \pm 1.2$ | $92.3 \pm 0.9$ | $71.87 \pm 0.17$ |
| **GRAND-nl** (ours) | $82.3 \pm 1.6$ | $70.9 \pm 1.0$ | $77.5 \pm 1.8$ | $\mathbf{92.4 \pm 0.3}$ | $82.4 \pm 2.1$ | $\mathbf{92.4 \pm 0.8}$ | $71.2 \pm 0.2$ |
| **GRAND-nl-rw** (ours) | $\mathbf{83.3 \pm 1.3}$ | $\mathbf{74.1 \pm 1.7}$ | $78.1 \pm 2.1$ | $91.3 \pm 0.7$ | $\mathbf{85.8 \pm 1.5}$ | $\mathbf{92.5 \pm 1.0}$ | $\mathbf{72.23 \pm 0.20}$ |

Table 1: Test accuracy and std for 20 random initializations and 100 random train-val-test splits. *Using labels. [†]using 1.5M parameters.

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
