# OpenReview forum: "GRAND: Graph Neural Diffusion"
_NeurIPS.cc/2021/Workshop/DLDE — DLDE Workshop -- NeurIPS 2021 Poster_

### Official Review · Reviewer_6Ya7 · 2021-10-11
**Novel idea unifying Graph Neural Networks and Neural PDEs with significant future impact**

**Confidence:** 4

**Review:**


The authors propose a novel framework that casts Graph Neural Networks as discretizations of the diffusion PDE on graphs, with a learned diffusivity function parameterized by an attention neural network. The idea is quite innovative and the work is quite valuable to the community, allowing future research on Graph Neural Networks to draw insights from the rich literature on computational methods for PDEs. The authors provide appropriate theoretical justifications for their model and the empirical evaluation is quite thorough.

I only have the following question: The authors draw a parallel between the time evolution of the diffusion operator with the number of layers in a GNN (lines 97-99). However, given the fact that the attention weights $\mathbf{W}_K$ and $\mathbf{W}_Q$ are chosen to be time-invariant, I'm not sure how strong the analogy is since different layers of a deep neural network generally use different parameters (this is different from iterative applications of the same operator, which, as far as I understand, is the case for time-domain discretization in GRAND). While this is in no way a limitation of the work, It would be great if the authors could add a comment on this, perhaps comparing with Neural ODE formalisms with time-varying parameters like ANODEV2.

[1]: Tianjun Zhang, Zhewei Yao, Amir Gholami, Joseph E. Gonzalez, Kurt Keutzer, Michael W. Mahoney, George Biros. ANODEV2: A Coupled Neural ODE Framework.  Advances in Neural Information Processing Systems, 2019.

**Score:**

4: Very good paper

---

### Official Review · Reviewer_dVKj · 2021-10-11
**Interesting work using PDEs to understand and design GNNs**

**Confidence:** 3

**Review:**

The paper introduces a class of graph neural networks derived from the discretized heat diffusion equation. The resulting GRAND model seems very promising and brings together insights from both differential equations and deep learning.  The paper is written clearly, and does a great job introducing the relevant concepts. Numerical results are looking strong, and showcase how flexible the suggested framework is.

The introduction states that “we show that many popular GNNs can be derived from a single framework[…]”, but this seems to be missing from the workshop-version of the manuscript, as do the promised studies of different time steppers and large layer depths. Please consider revising that second paragraph of the introduction.

Minor:

More details on numerical experiments would have been good (what values for dimensionality d_k, number of attention heads h etc.).

line 71: refers to Supplemental Materials that seem to be missing here.

line 119: I am not sure how the rewired edge set E’ (defined as subset of the original edge set ?) can include self-loops when the original edge set was assumed to be self-loop-free in line 56?

**Score:**

4: Very good paper

---

### Official Review · Reviewer_tMmM · 2021-10-14
**Interesting use of diffusion equations for GNNs**

**Confidence:** 3

**Review:**

The authors present a technique that uses a PDE-based diffusion model to treat Graph Neural Networks (GNNs).  The approach is able to address problems with GNNs, such as depth, over-smoothing, and bottlenecks.  The paper does a very good job of introducing diffusion equations PDE and how it applies more generally on manifolds.  The relationship to graphs and GNNs more specifically is then explained, with a detailed examination of the associated properties for applying the equations in the existing neural architecture.   The results are quite promising and generally fall on the upper end of the spectrum, beating state-of-the-art techniques on many benchmark tasks.  Caption should indicate coloring scheme, though the ranking of these is clear.  The fact that the GRAND-* family of schemes tend to out-perform all existing methods is impressive.  Claims made earlier in the paper are not corroborated in the evaluation, such as the depth, over-smoothing, and bottleneck issues of existing approaches.


**Score:**

4: Very good paper

---

### Decision · Program_Chairs · 2021-10-14

**Decision:**

Accept (Spotlight)

**Comment:**

The authors further explore connections between GNNs and differential equations and propose GRAND. All reviewers recommend acceptance.